# Field Prevalence and Pathological Features of *Edwardsiella tarda* Infection in Farmed American Bullfrogs (*Aquarana catesbeiana*)

**DOI:** 10.3390/ani15172487

**Published:** 2025-08-25

**Authors:** Yongping Ye, Yufang Huang, Furong Li, Ziyan Chen, Han Lin, Ruiai Chen

**Affiliations:** 1College of Veterinary Medicine, South China Agricultural University, Guangzhou 510642, China; yyp@stu.scau.edu.cn (Y.Y.); huangyufang@stu.scau.edu.cn (Y.H.); lfr@stu.scau.edu.cn (F.L.); czy2020czy@stu.scau.edu.cn (Z.C.); 2Zhaoqing Branch Center of Guangdong Laboratory for Lingnan Modern Agricultural Science and Technology, Zhaoqing 526238, China; 3Zhaoqing Dahuanong Biology Medicine Co., Ltd., Zhaoqing 526238, China

**Keywords:** *Edwardsiella tarda*, *Aquarana catesbeiana*, chronic enteritis, subclinical infection, infection dynamics

## Abstract

Bullfrogs are now widely farmed in China, yet hidden illnesses can quietly erode profits and animal welfare. In this study, we looked for the bacterium *Edwardsiella tarda* in apparently healthy bullfrogs from four farms in Guangdong Province. Laboratory tests showed that between two and three out of every five frogs carried the germ in their intestines, usually in low numbers. The strains we isolated already withstand some older medicines such as penicillin and tetracycline, though newer and more powerful antibiotics still work. When healthy frogs were exposed to the bacterium in the laboratory, very large doses killed them quickly, but much smaller doses were enough to cause long-lasting gut inflammation that slows growth. Because infected frogs often show no clear signs of disease, the pathogen can spread unnoticed in ponds and may become harder to treat if antibiotics are used incorrectly. Regular health checks, clean-water management and careful selection of medicines will help farmers keep their stock healthy, protect farm income, and reduce the risk that this organism will threaten other animals or people.

## 1. Introduction

*Edwardsiella tarda* is a Gram-negative, facultatively anaerobic, non-capsulated bacterium belonging to the order Enterobacterales and is recognised as a facultative intracellular pathogen [1,2,3]. Its host spectrum is broad, encompassing fish, amphibians, birds, mammals, and even humans [4,5]. In fish such as flounder and seahorse, *E*. *tarda* infection is chiefly characterised by enteritis accompanied by anorexia and intestinal fluid accumulation [6,7]. Reported avian cases, mostly involving ichthyophagous species, include septicaemia in crested ibis [8] and renal masses in common loons [9]. Consumption of raw or under-cooked fish contaminated with *E*. *tarda* can likewise lead to human fulminant septicaemia and become life-threatening [10], although human infection more commonly presents as self-limiting gastroenteritis that may progress to chronic diarrhoea [2,11]. Isolation of *E. tarda* has also been documented in pigs, puppies, and bullfrogs, yet the pathogenicity of these isolates has not been examined in detail [12,13,14].

The American bullfrog (*Aquarana catesbeiana*), native to North America, has become one of the fastest-growing aquaculture commodities in China. Driven by high economic returns, producers frequently adopt high-density stocking, which promotes water eutrophication and makes bacterial diseases difficult to control [15,16]. At present, meningitis-like disease caused by *Elizabethkingia miricola* is widespread [17,18], and red-leg syndrome (acute septicaemia) associated with *Aeromonas*, *Streptococcus*, and other bacteria attracts considerable attention because of the resulting mass mortalities [16,19]. In contrast, certain chronic conditions lack evident clinical signs and are therefore easily overlooked; surveys indicate that farmers primarily rely on sensory cues for routine monitoring [20]. Intensified farming increases animals’ susceptibility to chronic diseases and facilitates subclinical transmission of infectious agents [21,22]. Although such diseases may not cause immediate mortality, they markedly reduce production performance and profitability, while subclinically infected animals may continuously shed pathogens into the environment, greatly complicating control measures and amplifying the risk of spread [23,24].

Recently, we conducted an epidemiological survey across several bullfrog farms in Guangdong Province, China. During ongoing epidemiological surveillance, we identified multiple bullfrog populations that showed reduced feed intake but few overt clinical signs. The present study isolates and identifies the aetiological agent, delineates its epidemiological characteristics and disease progression, and thus aims to fill a knowledge gap concerning chronic infections in bullfrog culture while providing data useful for farm management.

## 2. Materials and Methods

### 2.1. Epidemiological Survey

Between 2022 and 2025, during a broad epidemiological survey, we identified a disease characterised by low mortality, minimal external symptoms, and reduced feed intake in farmed bullfrogs from Guangzhou, Zhaoqing, Huizhou, and Yunfu. For these asymptomatic cases, we typically collected 53 frogs from at least three randomly selected ponds per farm. Among them, 48 individuals were examined microscopically for parasites, and pooled liver, spleen, kidney, and intestinal tissues (50 mg total) were subjected to DNA extraction. The remaining DNA was stored at −20 °C for subsequent molecular analyses. The other five frogs were used for necropsy and bacterial isolation, and based on the bacterial isolation results, the DNA samples from the initial 48 individuals were reanalysed.

### 2.2. Experimental Animals

Healthy bullfrogs (20 ± 2 g) were obtained from a farm in Zhaoqing with no known history of disease. The frogs were maintained in a controlled laboratory environment at 25 ± 1 °C, with regular feeding and daily water replacement. All bullfrogs were humanely euthanised by cervical dislocation either prior to tissue collection or at the end of the experiments.

### 2.3. Histopathology

Liver and intestinal tissues were collected from both naturally infected bullfrogs (*n* = 5) and healthy individuals (*n* = 5). In the experimental infection model, intestinal samples were obtained from five frogs at each point (1, 3, 5, 9, and 14 days post-infection), with five uninfected frogs used as controls. All tissue samples were immersed in 4% paraformaldehyde for 96 h to ensure adequate fixation. Following fixation, the samples were dehydrated, cleared, and embedded in paraffin blocks using routine histological procedures. Sections were cut at a thickness of 3 to 5 μm, then stained with Mayer’s hematoxylin and eosin. Stained slides were examined under a light microscope (Eclipse Ci, Nikon, Tokyo, Japan) for histopathological assessment.

### 2.4. Bacterial Isolation and Identification

To isolate pathogenic bacteria, bullfrogs were dissected under sterile conditions, and samples were obtained from visibly affected liver, spleen, kidney, and intestinal tissues. The samples were inoculated onto brain heart infusion (BHI) agar and cultured at 30 °C for 18 h. Dominant bacterial colonies were selected and purified through repeated streaking. Genomic DNA was extracted from these purified isolates. Species identification was performed by amplifying and sequencing both the 16S rRNA and rpoB genes, following primer sets and PCR protocols previously established in our laboratory [25]. Sequencing was conducted by Tianyi Huayu Gene Technology Co., Ltd., Suzhou, China, and the resulting sequences were aligned with reference data in the NCBI database for preliminary classification. All bacterial strains were preserved at −80 °C for downstream analyses. One or two representative isolates from each site (designated E1, E2, E4, E5, E7, E8, and E9) were used for phylogenetic reconstruction using MEGA 7 software.

### 2.5. Quantitative Detection of FV3 and E. tarda in Field Samples

A total of 192 DNA samples preserved at −20 °C, as described in Section 2.1, were thawed on ice and subjected to qPCR detection and quantification of FV3 and *E. tarda* using previously published methods [26,27]. The target genes were the major capsid protein gene for FV3 and the gyrB gene for *E. tarda*. Primers and probe used in this assay are listed in Table 1. Samples with a cycle threshold value below 35 were considered positive. Based on the standard curves provided in the cited literature and the dilution factor introduced during DNA extraction, the viral or bacterial load in each sample was calculated (expressed as copies/μL for FV3 and CFU/μL for *E. tarda*). All reactions were performed in triplicate.

### 2.6. Antimicrobial Susceptibility Testing

Antimicrobial susceptibility of seven *E. tarda* isolates from bullfrogs was evaluated using the standard agar disc diffusion method according to the CLSI M100-Ed34 Performance Standards for Antimicrobial Susceptibility Testing [28]. A total of 18 antibiotics were tested, including Cefotaxime, Chloramphenicol, Streptomycin, Gentamicin, Imipenem, Tetracycline, and eleven additional agents to be supplemented. All antibiotic discs were obtained from Hangzhou Microbial Reagent Co., Ltd., Hangzhou, China. *Escherichia coli* ATCC 25,922 was used as the quality control strain and yielded inhibition zone diameters within the CLSI-recommended ranges. Based on the diameter of inhibition zones, isolates were classified as susceptible, intermediate, or resistant.

### 2.7. Detection of Virulence Genes

To determine virulence gene profiles, all *E. tarda* isolates were examined for eight genes previously reported as pathogenicity determinants in this species [29,30,31,32]. The target genes were *fliC* (flagellin gene), *ompA* (outer membrane protein A gene), *fimA* (fimbrial adhesin-like protein gene), *gadB* (glutamate decarboxylase gene), *katB* (catalase gene), *citC* (citrate lyase ligase gene), *mukF* (chromosome partitioning protein MukF gene), and *eseD* (type III secretion system effector protein D gene). PCR amplification followed protocols established in our earlier work. Products were resolved on 1 percent agarose gels and visualised with a gel imaging system (Bio-Rad, Berkeley, CA, USA). A clear band of the expected size was recorded as positive (“+”), whereas the absence of a band was recorded as negative (“−”).

### 2.8. Infection Model

#### 2.8.1. Challenge Experiment

*E. tarda* Strain E2 was cultured overnight at 28 °C in brain heart infusion broth. The resulting stationary-phase cells were serially diluted with sterile 0.65% physiological saline to prepare four inocula containing 1.2 × 10^9^, 1.2 × 10^8^, 1.2 × 10^7^, and 1.2 × 10^6^ CFU/mL, respectively. Groups of ten healthy bullfrogs were each given 0.2 mL of one dilution by intraperitoneal injection; the control group received an equivalent volume of sterile saline. All frogs were held in laboratory tanks at 25 ± 1 °C, and clinical signs together with mortality were recorded daily for fourteen consecutive days.

#### 2.8.2. Disease Progression Analysis

Fifty clinically healthy bullfrogs were allocated to the infection group and injected intraperitoneally with 200 µL of an *E. tarda* E2 suspension (1.2 × 10^8^ CFU/mL). A negative-control group of ten frogs received 200 µL of sterile physiological saline by intraperitoneal injection. At 1, 3, 5, 9, and 14 days post-injection, five frogs were randomly selected from each group. From every frog two intestinal samples were taken: one fragment was fixed in 4% paraformaldehyde for histopathology (Section 2.3), and the second fragment was pooled with those from the other four frogs sampled on the same day for gene-expression analysis (Section 2.9). All animals were maintained throughout the trial under the husbandry conditions described previously.

### 2.9. Intestinal Gene Expression

To investigate the impact of *E. tarda* infection on intestinal barrier integrity and mucosal inflammation in bullfrogs, we assessed the expression of key genes associated with epithelial tight junctions (*Claudin-7*, *Occludin*, and *ZO-2*) and pro-inflammatory signalling (*TNF*, *IL-8*, and *IL-1β*). These genes were selected based on their established roles in maintaining epithelial homeostasis and mediating acute immune responses in mucosal tissues [19,33]. The amphibian 18S rRNA gene was used as an internal reference [34]. Relative transcript levels were calculated using the 2^−ΔΔCT^ method, with the day-0 control group serving as the calibrator.

**Table 1 animals-15-02487-t001:** The information of primers used in this study.

Target Gene	Primer Sequence (5′-3′)	Reference
*IL-1β*	F: CAGTGATTAGACACCCAGGAACG	[19]
	R: GACCTTCATTTCAGTGGAGCATTC	
*IL-8*	F: GCTTGTGCCACCCTTACCCTC	[19]
	R: GCTCTACCCAAGAAGCAGAAGGA	
*TNF*	F: CACAGCACCCAGGAGCAACT	XM_040323112.1
	R: GGGATGATGTGGAAGTGGAGC	
*ZO-2*	F: TGAGGATGGACATTTTGACCGTAG	[33]
	R: CATAGCCTCGGTCTGGACTGGAT	
*Claudin-7*	F: GCTCTCCATTGTTCTCGGGGT	[33]
	R: CCCAGCAGAAAGATGAAGCCTC	
*Occludin*	F: GCTCTCCACCTGGCATCATCA	[33]
	R: GCAAAACCCATTCCCATCTGAG	
*RNA 18S*	F: CGTTGATTAAGTCCCTGCCCTT	[34]
	R: GCCGATCCGAGGACCTCACTA	
*16S rRNA*	F: AGAGTTTGATCCTGGTCAGAACGAACGCT	[25]
	R: TACGGCTACCTTGTTACGACTTCACCCC	
*gadB*	F: ATTTGGATTCCCGCTTTGGT	[31]
	R: GCACGACGCCGATGGTGTTC	
*mukF*	F: TTGCTGGCTATCGCTACCCT	[31]
	R: AACTCATCGCCGCCCTCTTC	
*citC*	F: TTTCCGTTTGTGAATCAGGTC	[31]
	R: AATGTTTCGGCATAGCGTTG	
*fimA*	F: CTGTGAGTGGTCAGGCAAGC	[31]
	R: TAACCGTGTTGGCGTAAGAGC	
*katB*	F: CTTAGCCATCAGCCCTTCC	[31]
	R: GCGAGTGCCGTAGTCCTT	
*EseD*	F: TTCAGGGTGGTCAGTATCTCG	AY643478.1
	R: TCAACAGACGCAGCAAAGC	
*OmpA*	F: ACCCGTCTGGACTATCAGTATGT	GQ259743.1
	R: GCGGCTGAGTAACTTCTTCTTT	
*FliC*	F: CGCTGATGGCACAGAATAAC	AP040123.1
	R: TCAGATAGGGCTTTGTCCAG	
*rpoB*	F: GCAGTGAAAGARTTCTTTGGTTC	[25]
	R: GTTGCATGTTNGNACCCA	
Et-*gyrB*	F: TGGCGACACCGAGCAGA	[26]
	R: ACAAACGCCTTAATCCCACC	
FV3-*MCP*	F: GACGAGAGACAGGCCATGAG	[27]
	R: GGAAGGGTGTGTGACGTTCT	
	Probe: ACCTCCTGATCCA	

### 2.10. PCR and Quantitative PCR

Primers and probes used in this study are listed in Table 1. Conventional PCR was performed to amplify the 16S rRNA and rpoB genes for bacterial identification (Section 2.4), as well as eight virulence genes described in Section 2.7. All reaction mixtures and thermal cycling conditions followed the protocol reported by Lin et al., 2023 [35].

Quantitative PCR assays for the detection of *E. tarda* and FV3 in field samples were carried out using the reagent systems and thermal cycling conditions described in previous studies [26,27]. Each reaction was performed in triplicate.

For intestinal gene expression analysis, total RNA was extracted from pooled intestinal samples collected at different time points (Section 2.8.2), reverse-transcribed to cDNA, and subjected to qPCR using the reagent system and thermal profile described by Lin et al., 2024 [19]. All reactions were performed in triplicate.

### 2.11. Statistical Analysis

Data are presented as the mean ± standard deviation from three independent trials. Prior to hypothesis testing, normal distribution was confirmed with the Kolmogorov–Smirnov test and equality of variances with the Levene test; both criteria were satisfied for every variable. Group differences were therefore evaluated with a single-factor analysis of variance. All statistical procedures were performed in SPSS version 26 (IBM, Armonk, NY, USA), and figures were prepared with GraphPad Prism version 10.4.1 (GraphPad Software, San Diego, CA, USA). Significance thresholds were defined as follows: ns for *p* > 0.05, * for *p* < 0.05, ** for *p* < 0.01 and *** for *p* < 0.001.

## 3. Results

### 3.1. Clinical Features

Gross findings are summarised in Figure 1. Diseased bullfrogs showed inflamed and oedematous intestines with dilated capillaries and occasional intussusception (Figure 1A). A subset of animals exhibited enlarged, pale livers (Figure 1B). Histologically, compared with healthy controls (Figure 2A,C), infected intestines displayed markedly shortened and disorganised villi, focal epithelial loss, pronounced submucosal oedema, and dense inflammatory cell infiltration (Figure 2B). Liver sections from the same animals revealed disrupted hepatic cords, hepatocellular swelling, and sinusoidal congestion (Figure 2D). These combined gross and microscopic lesions indicate that *E. tarda* infection compromises the intestinal barrier and initiates a systemic inflammatory response.

### 3.2. Molecular Identification

No parasites were detected in any of the sampled frogs, and FV3-specific qPCR was negative across all DNA extracts. A single bacterial morphotype was consistently recovered from the bacteriological cultures of liver, spleen, kidney, or intestinal tissues of sampled frogs. After purification the colonies were circular, smooth, white, and 0.5–1.0 mm in diameter, with no obvious haemolytic activity was observed on 5% sheep-blood agar (Appendix A). Sequences of *16S rRNA* and *rpoB* from seven representative isolates (E1, E2, E4, E5, E7, E8, E9) have been deposited in GenBank (accession numbers: PV915438–PV915444 for *16S rRNA*; PV935302–PV935308 for *rpoB*). BLAST analysis (performed using the NCBI online tool https://blast.ncbi.nlm.nih.gov/Blast.cgi, accessed in June 2025) showed > 99% identity to reference strains of *E. tarda*. Neighbour-joining phylogenies based on both loci grouped all seven isolates into a single, strongly supported *E. tarda* clade (Figure 3).

### 3.3. Prevalence and Bacterial Load

Using a previously published quantitative PCR method specific for *E. tarda*, all forty-eight DNA samples from each farm were re-examined. The detection rates in Guangzhou, Zhaoqing, Huizhou, and Yunfu were 45.8%, 39.6%, 77.1%, and 66.7%, respectively, indicating a high prevalence of the pathogen (Figure 4A). For positive specimens, most bacterial loads ranged from 10^2^ to 10^3^ CFU/µL (equivalent to 10^5^ to 10^6^ CFU/mL), reflecting generally low levels of colonisation across the four farms (Figure 4B).

### 3.4. Virulence Gene Profiling

PCR screening (Table 2) demonstrated that all seven *E. tarda* isolates (E1, E2, E4, E5, E7, E8, E9) possessed the core virulence genes *gadB*, *mukF*, *citC*, *fimA*, and *ompA*. Among the accessory loci examined, variation was confined to the flagellar gene *fliC*, which was detected in isolates E2 and E4 but absent from the other five strains. The type III secretion effector gene *eseD* and the oxidative stress gene *katB* were not amplified from any isolate. These findings indicate that the isolates share conserved metabolic and adhesion gene complements, and that flagellar motility potential is the only variable trait detected in the present study.

### 3.5. Antimicrobial Susceptibility Patterns

The antibiotic resistance of seven representative *E. tarda* strains isolated from four different regions of Guangdong Province was tested using the paper disc diffusion method, as shown in Table 3. The strains were generally resistant to trimethoprim, ampicillin, and tetracyclines. They were mostly susceptible to cephalosporins, fluoroquinolones, and carbapenems, although some strains (E1, E5) were resistant to ciprofloxacin. The aminoglycoside antibiotics showed variability: most strains were susceptible to amikacin and gentamicin, but exhibited intermediate resistance to kanamycin, with only strain E7 being susceptible. Overall, the prevalent strains exhibited multiple drug resistance, with certain regional differences observed among the strains.

### 3.6. Challenge Experiment

To confirm the pathogenicity of *E*. *tarda* in bullfrogs, strain E2 was administered at four inoculum levels. As shown in Figure 5, the highest dose of 1.2 × 10^9^ CFU/mL produced a cumulative mortality of 90% within fourteen days, with the peak loss recorded on day 4. At 1.2 × 10^8^ CFU/mL, only sporadic deaths occurred during the first week; the mortality peak shifted to the second week, and overall lethality declined. Survival after fourteen days reached 80–90% in groups inoculated with 1.2 × 10^7^ or 1.2 × 10^6^ CFU/mL. Bacteria re-isolated from the intestines of dead frogs were sequenced and confirmed as *E. tarda*. No deaths or lesions were observed in the saline control group. All remaining frogs were humanely euthanised at the end of the trial.

### 3.7. Disease Progression

To clarify the course of infection we combined sequential histology with quantitative PCR. Figure 6 shows that intestinal lesions progressed from mild epithelial degeneration with modest villus shortening on days 1 to 3, to pronounced submucosal oedema, vascular congestion, and focal haemorrhage on days 5 to 9, and culminated on day 14 in deep ulceration with partial loss of the intestinal wall. In concordance with these structural changes, expression of the tight-junction genes *Claudin-7*, *Occludin*, and *ZO-2* was already significantly lower than in healthy controls as early as day 1 post-infection and remained depressed thereafter (Figure 7A–C), indicating rapid and persistent barrier disruption. The pro-inflammatory transcripts *TNF*, *IL-8*, and *IL-1β* peaked between days 3 and 5 (Figure 7D–F). By day 14, *IL-8* and *IL-1β* had returned to baseline, whereas *TNF* remained significantly elevated (*p* < 0.05). Collectively, the histological and molecular data demonstrate that *E. tarda* infection causes an early loss of epithelial integrity, provokes an acute cytokine surge, and leads to sustained tissue injury despite partial attenuation of the inflammatory response.

## 4. Discussion

*E. tarda* infection is a recognised threat in fin-fish aquaculture, yet its pathogenic behaviour in farmed amphibians has received little systematic scrutiny. Through longitudinal surveillance across multiple farms, we show that *E. tarda* is widely endemic in Chinese bullfrog production, usually circulating as low-level, subclinical infection. By coupling field surveys with laboratory analyses we characterised the circulating isolates’ virulence repertoires, antimicrobial-resistance profiles, and disease-course features, thereby providing the first comprehensive baseline dataset for *E. tarda* infection in bullfrogs. These data address a critical knowledge gap for disease risk assessment and management in this rapidly expanding sector.

Field sampling revealed that most frogs harbour *E. tarda* at modest bacterial loads, a condition that nevertheless depresses growth and compromises immune competence through persistent intestinal inflammation. This chronic, largely silent presentation contrasts sharply with the fulminant septicaemia typically provoked by other bacterial agents of bullfrogs, such as *Aeromonas*, *Acinetobacter*, *Streptococcus*, and *Elizabethkingia*, which precipitate sudden mass mortalities [16,19,35,36]. Consistent with reports that *E. tarda* can sustain prolonged stimulation of host defences [37], our challenge model demonstrated that pathogenesis remains dose-dependent: when frogs received an inoculum of 1 × 10^9^ CFU/mL, cumulative mortality approached 90%, mirroring the acute septicaemic outcomes documented in infected fish species [38]. Although human cases rarely progress beyond self-limiting gastroenteritis, once bacteraemia is established the clinical course accelerates and case-fatality rates become alarmingly high [39]. Early recognition of subtle disease cues and an appreciation of the ensuing time-course are therefore indispensable for effective outbreak prevention and control in amphibian culture.

Infected hosts consistently developed progressive enteritis driven by a combination of direct bacterial invasion and dysregulated immune signalling. Experimental infection markedly elevated *IL-1β* and *IFN-γ* and thereby sustained mucosal inflammation that compromised intestinal barrier integrity [40]. Previous work indicates that *Edwardsiella* species can impair Th17-cell function, destabilise host–microbiota metabolic crosstalk and selectively enrich Proteobacteria, amplifying epithelial injury [41]. Parallel studies in teleosts have shown activation of the *MyD88/IRAK4/TAK1* cascade with over-expression of *IKKβ* and *IL-1β*, culminating in necrotic lesions of the intestine and liver [42], while loss of goblet cells and tight-junction integrity further increases intestinal permeability, locking the tissue into an inflammation–damage loop [40,43]. Human infections, though usually self-contained, may similarly masquerade as inflammatory-bowel-disease-like pathology during protracted inflammation [11]. Our bullfrog model reproduced these hallmarks: *IL-1β* and *IL-8* transcripts rose sharply then gradually declined as cytokine stores were exhausted, whereas the tight-junction proteins *Claudin-7*, *ZO-2*, and *Occludin* remained significantly down-regulated throughout infection. Histopathological examination corroborated the molecular findings, showing pronounced inflammatory-cell infiltration and progressive structural damage of the intestinal tissue. Collectively, the data underline a mechanistic convergence that spans fish, amphibian, and human hosts.

Differences in virulence-factor repertoires can translate into distinct pathogenic profiles among circulating strains. All bullfrog isolates in this study harbour *gadB*, *mukF*, *citC*, *fimA*, and *OmpA*, genes previously shown to enhance intracellular and acid-stress survival [44,45], accelerate proliferation [46], reinforce intestinal colonisation [47], and promote adhesion, invasion, and immune evasion [30]. Intriguingly, whereas fish-derived *E. tarda* rely on *katB* to withstand H_2_O_2_ and phagocyte-mediated killing [48,49], this gene was absent from every bullfrog isolate examined. Amphibian macrophages depend more on CSF1/IL34-regulated antioxidant pathways [50,51], and their oxidative-stress response varies across metamorphic stages [52,53]; teleosts, in contrast, employ a dual Keap1a/Keap1b system and robust *katG* up-regulation to counter oxidative pressure [54,55]. Such host-specific immune landscapes may relax the selective pressure for *katB* retention in amphibian-adapted strains. Alternatively, bullfrog isolates could compensate via redundant antioxidant systems, such as alkyl-hydroperoxide reductase (*Ahp*) or other peroxidases [56,57]. These hypotheses warrant further experimental validation.

An analogous pattern of host adaptation is evident in antimicrobial susceptibility. All bullfrog isolates were resistant to trimethoprim, ampicillin, and tetracyclines, which matches the widespread detection of trimethoprim and tetracycline residues in aquaculture ponds and sediments across South China [58,59] and corresponds to the continued environmental presence of *bla_TEM_* and related β lactamase genes in livestock and municipal wastewater despite the official prohibition of penicillin in fish farming since 2002 [60,61]. In sharp contrast to the rapidly rising resistance to cephalosporins and fluoroquinolones that has been reported for other aquaculture pathogens [62,63], our *E. tarda* isolates remained fully susceptible to third-generation cephalosporins, carbapenems and, apart from two strains, ciprofloxacin. This susceptibility may reflect limited direct exposure of *E. tarda* in bullfrog systems to these higher level agents, or an incomplete acquisition of transferable resistance determinants such as *qnr* or extended spectrum β lactamase genes that have already spread through bacterial communities in the Pearl River basin and other Chinese waters [64,65]. Sustained genomic surveillance that combines resistance genotyping with farm level audits of antimicrobial use is therefore critical for preventing the convergence of high virulence and broad drug resistance.

These interpretations are constrained by the limited number of isolates and farms included, which may not capture the full diversity of *E. tarda* in bullfrog production. In addition, the study did not include molecular analysis of resistance genes, which limits the resolution of resistance mechanisms. Future genomic studies are needed to explore the relationship between phenotypic resistance and resistance genes.

## 5. Conclusions

This study provides the first systematic evidence of subclinical *E. tarda* infection in farmed bullfrogs, integrating field detection, histopathology, virulence profiling, and antimicrobial resistance assessment. The results highlight a conserved core of virulence genes adapted to amphibian hosts and a distinct resistance pattern marked by reduced susceptibility to commonly used antibiotics but preserved sensitivity to critical drugs. These findings offer a valuable reference for disease monitoring and control in amphibian aquaculture and underscore the importance of continued surveillance and responsible antimicrobial stewardship to safeguard animal health and production sustainability.

## Figures and Tables

**Figure 1 animals-15-02487-f001:**
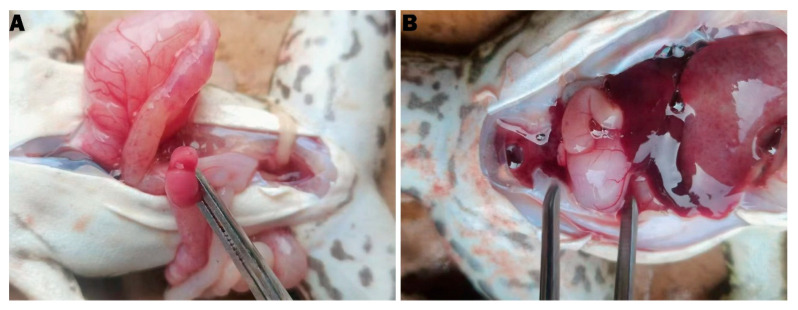
Gross lesions in naturally infected bullfrogs. (**A**) Intestine showing diffuse inflammation, pronounced oedema, dilated serosal capillaries, and an occasion of intussusception. (**B**) Liver from an affected frog with marked enlargement and a pale, haemorrhagic appearance indicative of blood loss.

**Figure 2 animals-15-02487-f002:**
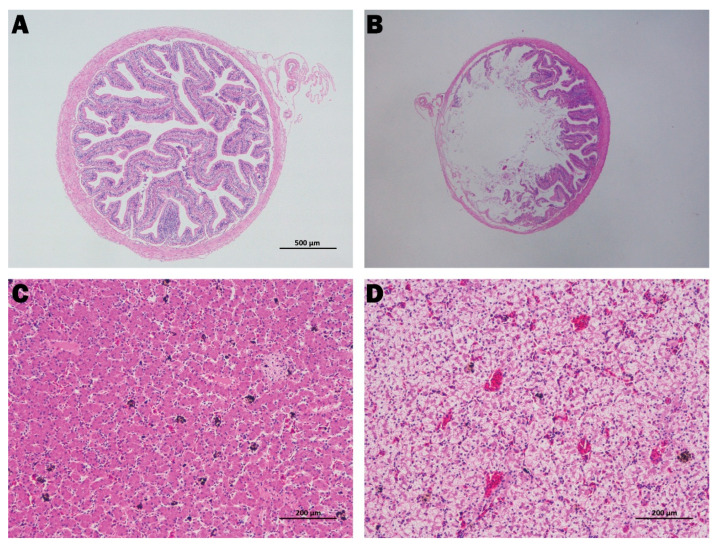
Histopathological alterations in intestine and liver. (**A**) Intestine of a healthy control frog displaying tall, regularly arranged villi and an intact epithelial lining. (**B**) Intestine of an infected frog with villus shortening, disorganisation, focal epithelial loss, pronounced submucosal oedema, and dense inflammatory infiltration. (**C**) Liver of a healthy control frog showing well-aligned hepatic cords and clear sinusoids. (**D**) Liver of an infected frog exhibiting disrupted hepatic cords, hepatocellular swelling, and sinusoidal congestion.

**Figure 3 animals-15-02487-f003:**
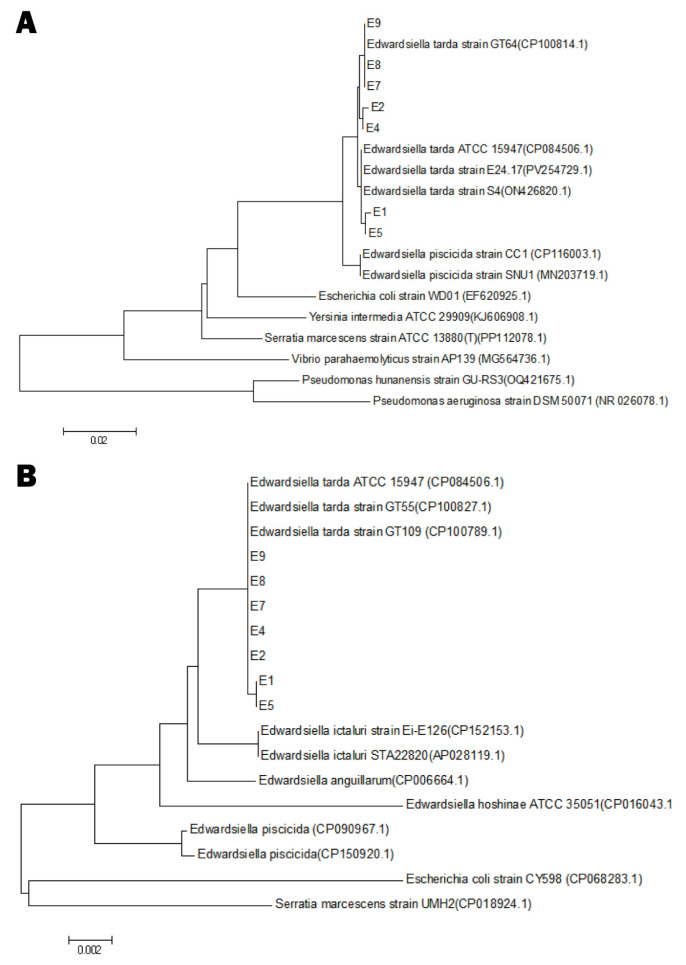
Phylogenetic analysis of seven bullfrog-derived *E. tarda* isolates. (**A**) Neighbour-joining tree inferred from partial *16S rRNA* sequences of isolates E1, E2, E4, E5, E7, E8, E9 together and reference strains. (**B**) Neighbour-joining tree based on the corresponding *rpoB* fragments from the same isolates and reference taxa. Both trees were generated in MEGA 7, with bootstrap values calculated from 1000 replicates. Scale bars indicate substitutions per nucleotide position.

**Figure 4 animals-15-02487-f004:**
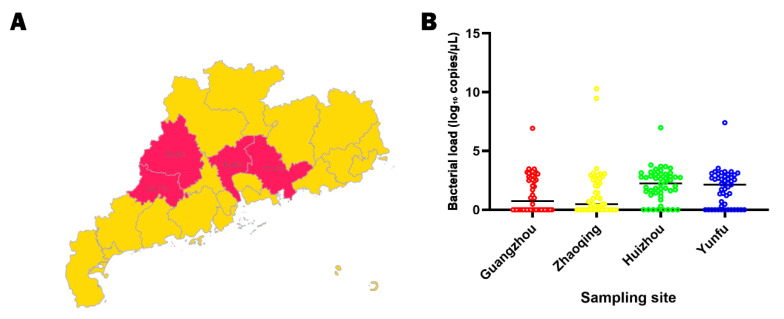
Spatial distribution and bacterial load of *E. tarda* in naturally infected bullfrogs. (**A**) Map of Guangdong Province indicating the locations of the four surveyed farms and the corresponding detection rates of *E*. *tarda* in each farm (*n* = 48 frogs per site). (**B**) Scatter plot showing bacterial loads (log_10_ CFU/µL) for all 48 frogs sampled at each site; horizontal lines represent the median for each farm.

**Figure 5 animals-15-02487-f005:**
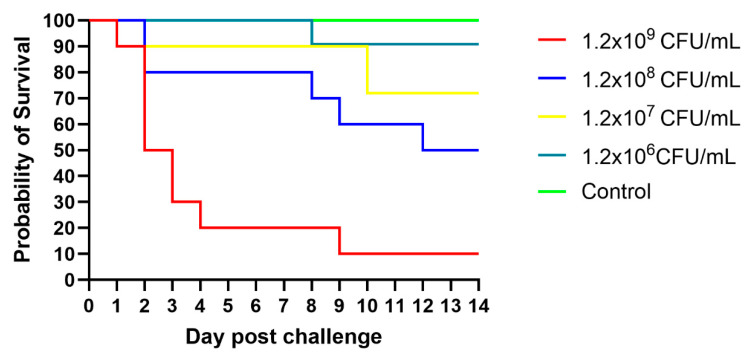
Survival of bullfrogs challenged with *E. tarda* strain E2.

**Figure 6 animals-15-02487-f006:**
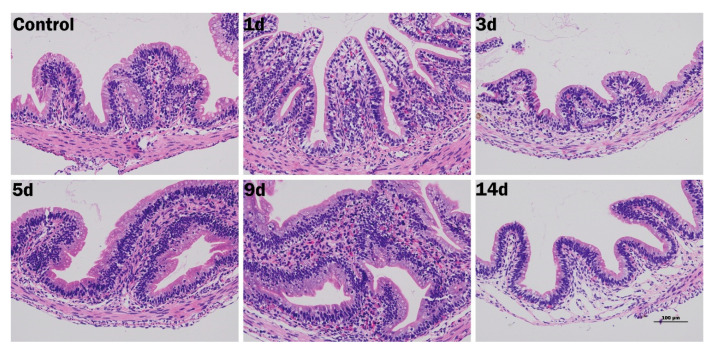
Histological progression of bullfrog intestine after *E. tarda* infection. Haematoxylin and eosin micrographs are presented, from left to right, for the healthy control and for samples collected at 1, 3, 5, 9, and 14 days post-infection. The control tissue shows intact villi with no inflammatory infiltrate. Mild epithelial degeneration and slight villus shortening are visible at days 1 and 3. By day 5, pronounced submucosal oedema and vascular congestion are evident, becoming more severe with focal haemorrhage and dense leucocyte infiltration at day 9. Deep ulceration and segmental loss of the intestinal wall are observed at day 14. Scale bar, 100 µm.

**Figure 7 animals-15-02487-f007:**
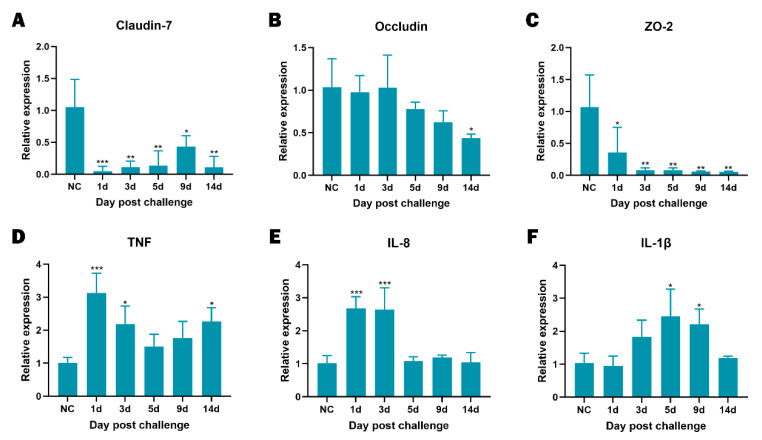
Temporal expression of intestinal genes in bullfrogs after *E. tarda* infection. Relative mRNA levels of (**A**) Claudin-7, (**B**) Occludin, (**C**) ZO-2, (**D**) TNF, (**E**) IL-8, and (**F**) IL-1β were determined by quantitative PCR at 0, 1, 3, 5, 9, and 14 days post-infection. Differences among time points were evaluated by one-way ANOVA. Significance relative to the day-0 control (NC) is indicated as follows: *, *p* < 0.05; **, *p* < 0.01; ***, *p* < 0.001.

**Table 2 animals-15-02487-t002:** Virulence gene profiling of 7 *E. tarda* strains.

Strains	Virulence Factor Genes							
	gadB	mukF	citC	fimA	katB	eseD	ompA	fliC
E1	+	+	+	+	−	−	+	−
E2	+	+	+	+	−	−	+	+
E4	+	+	+	+	−	−	+	+
E5	+	+	+	+	−	−	+	−
E7	+	+	+	+	−	−	+	−
E8	+	+	+	+	−	−	+	−
E9	+	+	+	+	−	−	+	−

Note: “+”, positive; “−”, negative.

**Table 3 animals-15-02487-t003:** Antibiotic susceptibility patterns of 7 *E. tarda* strains to 18 antimicrobial agents.

Classifications	Antibiotics	Content (µg)	Criteria (mm)				Isolation Sites/Isolated Strains/Inhibition Zone (mm)							
							Huizhou		Zhaoqing		Guangzhou		Yunfu	ATCC 25922 (CLSI Range)
			R	I	SDD	S	E1	E5	E2	E4	E7	E8	E9	
Cephalosporins	Ceftazidim	30	≤17	18–20	-	≥21	25 ^S^	25 ^S^	22 ^S^	21 ^S^	27 ^S^	27 ^S^	28 ^S^	27 (25–32)
	Cefepime	30	≤18	-	19–24	≥25	26 ^S^	24 ^SDD^	28 ^S^	28 ^S^	28 ^S^	28 ^S^	30 ^S^	36 (31–37)
Fluoroquinolones	Ciprofloxacin	5	≤21	22–25	-	≥26	16^ R^	18 ^R^	28 ^S^	28 ^S^	29 ^S^	29 ^S^	30 ^S^	29 (29–38)
	Ofloxacin	5	≤12	13–15	-	≥16	22 ^S^	20 ^S^	21 ^S^	22 ^S^	23 ^S^	20 ^S^	22 ^S^	30 (29–33)
Chloramphenicol	Chloramphenicol	30	≤12	13–17	-	≥18	13 ^I^	16 ^I^	15 ^I^	16 ^I^	18 ^S^	16 ^I^	16 ^I^	25 (21–27)
Aminoglycosides	Streptomycin	10	≤11	12–14		≥15	18 ^S^	14 ^I^	18 ^S^	18 ^S^	16 ^S^	18 ^S^	16 ^S^	18 (12–20)
	Amikacin	30	≤16	17–19	-	≥20	25 ^S^	21 ^S^	24 ^S^	22 ^S^	23 ^S^	22 ^S^	26 ^S^	21 (19–26)
	Gentamicin	10	≤14	15–17		≥18	24 ^S^	22 ^S^	24 ^S^	24 ^S^	24 ^S^	26 ^S^	22 ^S^	22 (19–26)
	Kanamycin	30	≤13	14–17	-	≥18	14 ^I^	14 ^I^	16 ^I^	15 ^I^	18 ^S^	17 ^I^	17 ^I^	18 (17–25)
Penicillin	Ampicillin	10	≤13	14–16	-	≥17	0 ^R^	0 ^R^	0 ^R^	0 ^R^	0 ^R^	0 ^R^	0 ^R^	17 (15–22)
Carbapenem	Imipenem	10	≤19	20–22	-	≥23	26 ^S^	30 ^S^	28 ^S^	28 ^S^	30 ^S^	28 ^S^	28 ^S^	28 (26–32)
Tetracycline	Tetracycline	30	≤11	12–14	-	≥15	10 ^R^	15 ^S^	8 ^R^	8 ^R^	18 ^S^	22 ^S^	18 ^S^	23 (18–25)
	Doxycycline	30	≤10	11–13	-	≥14	10 ^R^	11 ^I^	10 ^R^	12 ^I^	10 ^R^	10 ^R^	11 ^R^	21 (18–24)
	Minocycline	30	≤12	13–15	-	≥16	7 ^R^	10 ^R^	9 ^R^	9 ^R^	8 ^R^	9 ^R^	10 ^R^	20 (19–25)
DHFR inhibitor	Trimethoprim	5	≤10	11–15	-	≥16	0 ^R^	0 ^R^	0 ^R^	0 ^R^	0 ^R^	0 ^R^	0 ^R^	22 (21–28)

Note: ^S^, susceptible. ^I^, intermediate. ^R^, resistant. ^SDD^, Susceptible-Dose Dependent. DHFR inhibitor, Dihydrofolate reductase inhibitor.

## Data Availability

The original contributions presented in this study are included in the article. Further inquiries can be directed to the corresponding author.

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
