# Peer review of "Field Prevalence and Pathological Features of Edwardsiella tarda Infection in Farmed American Bullfrogs (Aquarana catesbeiana)"

_animals, 2025, doi:10.3390/ani15172487_

Round 1

Reviewer 1 Report

Comments and Suggestions for Authors

Line 22:              The statement „if antibiotics are misused” may not be entirely correct. It may also happen if the wrong antimicrobial agent is used, thereby supporting the need for antimicrobial susceptibility testing before antimicrobial agents are used.

Line 26:              Seven strains is a small number, maybe too small to draw general conclusions.

Line 29:              rpoB in italics

Line 34:              “penicillins” is mentioned, but only one agent of this class, the aminopenicillin ampicillin has been tested.

Lines 50, 54:      E. in E. tarda also in italics

Line 57:              E. tarda in italics

Line 64:              Aeromonas, Streptococcus in italics

Line 81ff            In Materials and Methods, there is an ethics statement missing for the animal experiments conducted. This must be obtained prior to the experiments. Without an ethics statement by the respective authorities, the results of a study like this cannot be published. There is also no information about how the bullfrogs were killed.

Lines 129-137:      Please delete the reference from 1966 Kirby and Bauer - this reference is by far outdated. Instead, please refer to the agar disk diffusion method as outlined in the CLSI documents. The authors need to provide the disk load in µg for each antimicrobial disk used. Moreover, there is no quality control strain mentioned, which has been tested side-by-side with the test strains. Without quality controls, results of antimicrobial susceptibility testing cannot be regarded as valid. Finally, the correct term is “susceptible”, not “sensitive”.

Lines 201-208:      Delete the “.” after Figure  and replace it by a blank.

Line 228 ff:            The statement “A single bacterial morphotype was consistently isolated from liver, spleen, kidney, and intestinal tissues” must be taken with care. As it stands, it sounds that all examined bullfrogs have suffered from a generalized infection (sepsis) that includes liver, spleen, kidneys and the intestine. However, in Materials and Methods, line 88, the authors described that they used “pooled liver, spleen, kidney, and intestinal tissues (50 mg total)”. This should be rephrased as the detection of E. tarda might originate from the intestinal tissue in these bulk samples.

Table 2:                   In Table 2 and in the associated text, please make sure that you write genes in lower case letters and in italics - actually, some are written with a capital letter. I also suggest to change the heading of this Table from Virulence factors to “Virulence factor genes” as you did not identify the virulence factors, but the respective genes.

Line 271:                 This heading should read “Antimicrobial susceptibility patterns”

Lines 274, 386:     Please replace penicillin by ampicillin. Enterobacteriaceae are commonly intrinsically resistant to penicillin, but not to aminopenicillins, such as ampicillin. Ampicillin was the agent that was tested.

Table 3:                   Quinolone must read “Fluoroquinolones”. There are no CLSI-approved clinical breakpoints for miloxacin. I do not know where the clinical breakpoints indicated in Table 3 come from. Please delete the information about miloxacin. In the footnote to Table 3, please replace sensitive by susceptible. Colistin B is mentioned in line 132, but no data are included in Table 3

Author Response

Comment 1: Line 22: The statement „if antibiotics are misused” may not be entirely correct. It may also happen if the wrong antimicrobial agent is used, thereby supporting the need for antimicrobial susceptibility testing before antimicrobial agents are used.

Response: Thank you for the suggestion. We have revised “if antibiotics are misused” to “if antibiotics are used incorrectly” to encompass both the misuse of antibiotics and the use of an inappropriate antimicrobial agent, while keeping the wording concise and accessible for general readers in the Simple Summary.

Comment 2: Line 26:Seven strains is a small number, maybe too small to draw general conclusions.

Response: We appreciate the reviewer’s concern. We will address this point in the Discussion by noting that the limited number of strains is a study limitation. In addition, we would like to clarify that our strain selection was intentional: preliminary screening in our laboratory showed that isolates from the same farm or nearby farms often exhibited nearly identical antimicrobial resistance profiles. To avoid redundancy and potential bias from repeated clones, we chose representative isolates from each site for detailed characterisation.

Comment 3: Line 29:              rpoB in italicsï¼›Lines 50, 54:      E. in E. tarda also in italicsï¼›Line 57:              E. tarda in italicsï¼›Line 64:              Aeromonas, Streptococcus in italics.

Response: We appreciate the reviewer’s careful reading and apologise for these oversights. All instances mentioned have been corrected, and we have checked the entire manuscript to ensure consistent use of italics for gene names and scientific names.

Comment 4: penicillins” is mentioned, but only one agent of this class, the aminopenicillin ampicillin has been tested.

Response: We appreciate the reviewer’s comment. Penicillins are rarely used in aquaculture, and our testing of the aminopenicillin ampicillin was intended to characterise the resistance profile of the isolates rather than to inform treatment choices. According to internationally accepted practice in antimicrobial susceptibility reporting, resistance or intermediate resistance to one representative agent of a given antimicrobial class can be recognised as resistance to that class. On this basis, resistance to ampicillin was described in the Abstract as resistance to penicillins.

Comment 5:   Line 81 In Materials and Methods, there is an ethics statement missing for the animal experiments conducted. This must be obtained prior to the experiments. Without an ethics statement by the respective authorities, the results of a study like this cannot be published. There is also no information about how the bullfrogs were killed.

Response: We appreciate the reviewer’s concern. In accordance with the journal’s template, the ethics statement is provided at the end of the manuscript (Lines 425–428). A scanned copy of the original IACUC approval and the qualification documents of the issuing institution have already been sent to the Editor via email for internal verification. As these documents contain other unpublished experimental details, we have not included them in the manuscript. In addition, we have revised the Materials and Methods section to specify the euthanasia procedure.

Comment 6: Lines 129-137:      Please delete the reference from 1966 Kirby and Bauer - this reference is by far outdated. Instead, please refer to the agar disk diffusion method as outlined in the CLSI documents. The authors need to provide the disk load in µg for each antimicrobial disk used. Moreover, there is no quality control strain mentioned, which has been tested side-by-side with the test strains. Without quality controls, results of antimicrobial susceptibility testing cannot be regarded as valid. Finally, the correct term is “susceptible”, not “sensitive”.

Response: We appreciate the reviewer’s constructive comments. We have re-performed the antimicrobial susceptibility tests in strict accordance with the CLSI M100-Ed34 guidelines. Escherichia coli ATCC 25922 was used as the quality control strain and showed inhibition zone diameters within the CLSI-recommended ranges. The concentrations (µg) of each antibiotic disk have now been added to the results table. In addition, we have replaced the term “sensitive” with the correct term “susceptible” throughout the manuscript.

Comment 7: Lines 201-208:      Delete the “.” after Figure  and replace it by a blank.

Response: We apologise for this oversight and thank the reviewer for pointing it out. We have deleted the unnecessary period after “Figure” and carefully checked the entire manuscript to ensure consistency in formatting.

Comment 8: Line 228 ff:            The statement “A single bacterial morphotype was consistently isolated from liver, spleen, kidney, and intestinal tissues” must be taken with care. As it stands, it sounds that all examined bullfrogs have suffered from a generalized infection (sepsis) that includes liver, spleen, kidneys and the intestine. However, in Materials and Methods, line 88, the authors described that they used “pooled liver, spleen, kidney, and intestinal tissues (50 mg total)”. This should be rephrased as the detection of E. tarda might originate from the intestinal tissue in these bulk samples.

Response: We thank the reviewer for this important clarification. The epidemiological survey (Methods, Line 88) was designed only to estimate the carriage rate of E. tarda in apparently healthy frogs, using pooled tissue samples, and not to determine the precise organ of origin. In contrast, the phylogenetic analysis section refers to isolates obtained from bacteriological culture. To avoid any misunderstanding, we have revised the sentence to read: “A single bacterial morphotype was consistently recovered from the bacteriological cultures of liver, spleen, kidney, or intestinal tissues of sampled frogs.” This wording clarifies that the isolates were obtained from organ cultures, without implying that all frogs suffered from systemic infection.

Comment 9: Table 2:                   In Table 2 and in the associated text, please make sure that you write genes in lower case letters and in italics - actually, some are written with a capital letter. I also suggest to change the heading of this Table from Virulence factors to “Virulence factor genes” as you did not identify the virulence factors, but the respective genes.

Response: We thank the reviewer for this careful observation. We have corrected all gene names in Table 2 and in the associated text to lower case italics to follow the standard nomenclature. In addition, we have revised the table heading from “Virulence factors” to “Virulence factor genes” and checked the entire manuscript to ensure that all related expressions have been modified consistently.

Comment 10: Line 271:                 This heading should read “Antimicrobial susceptibility patterns”

Response: We thank the reviewer for the suggestion. We have revised the heading of Section 3.5 to "Antimicrobial susceptibility patterns" to better reflect the content and align with standard terminology.

Comment 11: Lines 274, 386:     Please replace penicillin by ampicillin. Enterobacteriaceae are commonly intrinsically resistant to penicillin, but not to aminopenicillins, such as ampicillin. Ampicillin was the agent that was tested.

Response: We greatly appreciate the reviewer’s comments. Firstly, as mentioned in our previous response, penicillins are banned in aquaculture in China, and we selected ampicillin as a representative aminopenicillin solely to assess the resistance characteristics of the isolates. Secondly, since there is no quality control range for penicillin for E. coli ATCC 25922 in CLSI guidelines, and following standard practice, we chose a drug for which the quality control strain is known to be susceptible. For this reason, we still consider ampicillin to be the most appropriate choice.

We hope this clarifies our rationale, and we are happy to discuss any further concerns the reviewer may have.

Comment 12: Quinolone must read “Fluoroquinolones”. There are no CLSI-approved clinical breakpoints for miloxacin. I do not know where the clinical breakpoints indicated in Table 3 come from. Please delete the information about miloxacin. In the footnote to Table 3, please replace sensitive by susceptible. Colistin B is mentioned in line 132, but no data are included in Table 3.

Response: We appreciate the reviewer’s valuable comments.

  • We have replaced “Quinolone” with “Fluoroquinolones” in Table 3 to accurately reflect the antibiotic class.
  • We have corrected the term “miloxacin” in Table 3, which was a typographical error, and it should refer to Minocycline.
  • We have removed the reference to “Colistin B” from Line 132 as there is no corresponding data in Table 3.
  • In the footnote to Table 3, we have replaced “sensitive” with “susceptible” as per standard terminology in antimicrobial susceptibility testing.

Reviewer 2 Report

Comments and Suggestions for Authors

The manuscript entitled “Field prevalence and pathological features of Edwardsiella tarda infection in farmed American bullfrogs(Aquarana catesbeiana)” is an interesting investigation highlighting the role of this food-bred frog as a carrier of a pathogen with considerable zoonotic potential and worthy of attention. Another noteworthy aspect is the prevalence of strains resistant to several antibiotics commonly used in therapy, which is a cause for concern in light of the current emergency situation. A note of merit for the authors is therefore already to have drawn attention to a topical subject that has been little investigated in recent years.

Precisely, in relation to this aspect of resistance, however, it would have been interesting to have an overview of the resistance genes for the most important classes of antibiotics in order to be able to make considerations regarding phenotypic and genotypic resistance. I understand, however, that the authors preferred to investigate virulence factors, but this additional investigation would make the work complete and would be an important added value.

There is nothing to object about the experimental phase, especially the rigour with which the challenge test was designed and conducted.

I find myself at a loss because from my point of view the absence of molecular analysis of resistance genes is a major deficiency, but as far as the authors' intended target is concerned this shortcoming can safely be considered a minor one.

For this reason, I consider the manuscript to be publishable without any particular changes or corrections except, if possible, the integration of this molecular part, which would obviously enrich and modify part of the discussion and conclusions.

Author Response

Comment 1: The manuscript entitled Field prevalence and pathological features of Edwardsiella tarda infection in farmed American bullfrogs(Aquarana catesbeiana) is an interesting investigation highlighting the role of this food-bred frog as a carrier of a pathogen with considerable zoonotic potential and worthy of attention. Another noteworthy aspect is the prevalence of strains resistant to several antibiotics commonly used in therapy, which is a cause for concern in light of the current emergency situation. A note of merit for the authors is therefore already to have drawn attention to a topical subject that has been little investigated in recent years.

Precisely, in relation to this aspect of resistance, however, it would have been interesting to have an overview of the resistance genes for the most important classes of antibiotics in order to be able to make considerations regarding phenotypic and genotypic resistance. I understand, however, that the authors preferred to investigate virulence factors, but this additional investigation would make the work complete and would be an important added value.

There is nothing to object about the experimental phase, especially the rigour with which the challenge test was designed and conducted.

I find myself at a loss because from my point of view the absence of molecular analysis of resistance genes is a major deficiency, but as far as the authors' intended target is concerned this shortcoming can safely be considered a minor one.

For this reason, I consider the manuscript to be publishable without any particular changes or corrections except, if possible, the integration of this molecular part, which would obviously enrich and modify part of the discussion and conclusions.

Response: We greatly appreciate the reviewer’s thoughtful comments and valuable suggestion regarding the inclusion of resistance gene analysis. We fully agree that incorporating molecular data on resistance genes would enhance the study and provide important insights into the relationship between genotypic and phenotypic resistance.

However, we are currently conducting a larger-scale epidemiological survey focused on the prevalence, transmission, and relationship between resistance genes and antimicrobial resistance. This ongoing investigation will be the subject of a future publication, where we will address these aspects in greater detail. Therefore, we have decided not to include the molecular analysis of resistance genes in this particular study.

We have, however, acknowledged this limitation in the discussion section, emphasizing that this aspect will be further explored in future research.

Once again, we sincerely thank the reviewer for their constructive feedback, which has helped us improve the clarity and rigor of our study.

Reviewer 3 Report

Comments and Suggestions for Authors

The article is very well written and interesting. The topic is certainly of interest to the aquaculture sector. The authors designed and conducted the experimental sections correctly and well-structured. I have nothing particular to report to the authors; conceptually, the arguments are clear. I have highlighted some typos in the attached PDF that need to be corrected.

Author Response

Comment 1: Comments and Suggestions for Authors

The article is very well written and interesting. The topic is certainly of interest to the aquaculture sector. The authors designed and conducted the experimental sections correctly and well-structured. I have nothing particular to report to the authors; conceptually, the arguments are clear. I have highlighted some typos in the attached PDF that need to be corrected.

Response: We sincerely thank you for your positive feedback and for highlighting the typos in the attached PDF. We have carefully reviewed the document and made the necessary corrections to address all the identified spelling issues. We appreciate your valuable input and believe these changes have improved the clarity and quality of the manuscript.

Comment 2: Line 3: Move the bracket closer to catesbeiana, please.

Response: Thank you for your suggestion. We have made the requested change and moved the bracket closer to catesbeiana as per your recommendation.

Comment 3: Line 50,54,57: Please put everything in italics.

Response: Thank you for your comment. We have made the necessary revisions and ensured that everything is now in italics as requested.

Comment 4: Line 112,123: Please italicize the gene name.

Response: Thank you for your comment. We have revised the manuscript to ensure that all gene names are italicized when referenced individually. When referring to the gene name followed by the word "gene," both are not italicized, in accordance with standard nomenclature guidelines.

Comment 5:   Lines 159-162 The E. tarda suspension was inoculated intraperitoneally. The saline solution, however, was administered intramuscularly. I'm not entirely clear on the rationale for this different inoculation route.

Response: Thank you for your comment. We apologize for the confusion; this was a typographical error. In fact, all inoculations were performed intraperitoneally, including both the E. tarda suspension and the saline solution. We have corrected the manuscript to reflect this and ensure consistency.

Comment 6: Line230:     Separate "with" from "no", please.

Response: Thank you for your suggestion. We have separated "with" from "no" as requested and made the necessary revision in the manuscript.

Comment 7: Line 238:     "isolates" NOT in italics, please.

Response: Thank you for your comment. We have corrected the manuscript and ensured that "isolates" is not in italics, as per your suggestion.

Comment 8: Line 367:            After the dot, capitalize i please.

Response: Thank you for your suggestion. We have corrected the manuscript and capitalized the "i" after the dot as requested.

Comment 9: Line 389:           Please italicize the gene name.

Response: Thank you for your comment. We have italicized the gene name as requested.

Round 2

Reviewer 1 Report

Comments and Suggestions for Authors

Line 35, 279/280, 392:   “penicillins” is mentioned, but only one agent of this class, the aminopenicillin ampicillin has been tested. Thus, the term “penicillins” is too brad, especially since E. tarda exhibits intrinsic resistance to members of this class (e.g. penicillin G). Please replace penicillins by ampicillin.

The response “According to internationally accepted practice in antimicrobial susceptibility reporting, resistance or intermediate resistance to one representative agent of a given antimicrobial class can be recognised as resistance to that class.” is not entirely correct. This works only for antimicrobial classes in which a class representative is defined. However, sometimes, it also depends on the resistance mechanisms present. For example, if chloramphenicol is tested and an isolate proves to be chloramphenicol-resistant, it is not automatically also florfenicol-resistant.

Table 3:          Aminoglycosides; insert a space between zone and (mm); Table 2 is at the wrong place in the manuscript – the Table should appear immediately after the Table heading … and not after the heading 3.6. Challenge experiment.

Line 279:        trimethoprim with a lower case t

Line 281:        ciprofloxacin with a lower case c

Line 373:        Histopathological

Line 408-409:     In addition instead of Additionally.

Author Response

Comment 1: Line 35, 279/280, 392:   “penicillins” is mentioned, but only one agent of this class, the aminopenicillin ampicillin has been tested. Thus, the term “penicillins” is too brad, especially since E. tarda exhibits intrinsic resistance to members of this class (e.g. penicillin G). Please replace penicillins by ampicillin.

The response “According to internationally accepted practice in antimicrobial susceptibility reporting, resistance or intermediate resistance to one representative agent of a given antimicrobial class can be recognised as resistance to that class.” is not entirely correct. This works only for antimicrobial classes in which a class representative is defined. However, sometimes, it also depends on the resistance mechanisms present. For example, if chloramphenicol is tested and an isolate proves to be chloramphenicol-resistant, it is not automatically also florfenicol-resistant.

Response: We sincerely thank the reviewer for the very careful and constructive comments. We fully agree with the concern regarding the use of the term “penicillins.” As only ampicillin was tested in our study, we have replaced “penicillins” with “ampicillin” at all relevant places in the manuscript (Abstract, Results, and Discussion).

Regarding the reviewer’s additional clarification, we apologize for not having expressed our previous response clearly. What we intended to convey is not that resistance to one drug automatically implies resistance to all other members of the same class. Rather, according to internationally accepted definitions, if an isolate shows resistance or intermediate resistance to at least one representative agent of a given antimicrobial class, it can be designated as resistant to that class as a whole. For example, an isolate resistant to imipenem but susceptible to other carbapenems would still be categorized as “carbapenem-resistant” according to the CDC 2015 CRE definition and CLSI M100 guidelines. We have revised our wording to avoid ambiguity and better align with the reviewer’s valuable advice.

Comment 2: Table 3:          Aminoglycosides; insert a space between zone and (mm); Table 2 is at the wrong place in the manuscript – the Table should appear immediately after the Table heading … and not after the heading 3.6. Challenge experiment.

Response: We thank the reviewer for pointing out these formatting issues. We have corrected the terminology and spacing in Table 3 (inserting a space between “zone” and “(mm)”) and have also adjusted the table placement as suggested.

Comment 3: Line 279:        trimethoprim with a lower case t;Line 281:        ciprofloxacin with a lower case c.

Response: We thank the reviewer for the careful attention to detail. We have corrected the capitalization and now use lower case for trimethoprim (Line 279) and ciprofloxacin (Line 281) in the revised manuscript.

Comment 4: Line 373:        Histopathological.

Response: We thank the reviewer for noticing this spelling error. The word has been corrected from Istopathological to Histopathological (Line 373) in the revised manuscript.

.

Comment 5:   Line 408-409: In addition instead of Additionally.

Response: We thank the reviewer for this helpful suggestion. We have replaced Additionally with In addition at Lines 408–409 in the revised manuscript.